# Dysregulated Hippo Signaling Pathway and YAP Activation in Atopic Dermatitis: Insights from Clinical and Animal Studies

**DOI:** 10.3390/ijms242417322

**Published:** 2023-12-10

**Authors:** Ga Hee Jeong, Ji Hyun Lee

**Affiliations:** 1Department of Biomedicine & Health Sciences, The Catholic University of Korea, #222 Banpo-daero, Seocho-gu, Seoul 06591, Republic of Korea; worldgh27@catholic.ac.kr; 2Department of Dermatology, Seoul St. Mary’s Hospital, College of Medicine, The Catholic University of Korea, #222 Banpo-daero, Seocho-gu, Seoul 06591, Republic of Korea

**Keywords:** atopic dermatitis, hippo pathway, yes-associated protein (YAP), transcription activator with a PDZ-binding motif (TAZ), JAK–STAT signaling

## Abstract

The yes-associated protein (YAP) of the Hippo pathway regulates a variety of target genes involved in cell proliferation, survival, and inflammation. YAP and transcription activator with a PDZ-binding motif (TAZ) proteins act as mediators of the inflammatory response. Still, their role in atopic dermatitis (AD)—particularly, the association with the nuclear factor kappa-B and Janus kinase (JAK)-signal transducer and activator of transcription (STAT) pathways—is not fully understood. In this study, we found that YAP, is upregulated in AD patients and NC/Nga mouse model of AD. In addition, inhibition of YAP significantly reduced epidermal cell proliferation by 58% and mast cell numbers by 51% and attenuated the upregulation of both Th1- and Th2-associated cytokines. Among the JAK-STAT family proteins, the expressions of JAK1 and JAK2 and those of STAT1, STAT2, and STAT3 were also downregulated. These findings may explain the role of YAP in AD and suggest YAP inhibitors as promising therapeutic agents for AD.

## 1. Introduction

Atopic dermatitis (AD) is a prevalent and persistent inflammatory skin disease characterized by a complex interplay of genetic, environmental, and immunological factors. This interplay results in impaired skin barrier function and inflammation accompanied by itching. Despite remarkable advances in our understanding of AD, there are still patients with uncontrolled, refractory AD. Moreover, current treatments have some side effects [1]; for example, the leading calcineurin inhibitor, tacrolimus, can cause side effects such as a burning sensation, while the biologic dupilumab has been linked with conjunctivitis and paradoxical head and neck erythema [2,3]. Recently, several Janus kinase (JAK) inhibitors have been introduced as treatments for refractory AD; however, baricitinib can cause nasopharyngitis, folliculitis, and herpes infection; abrocitinib can trigger headache and nausea; and upadacitinib can cause acne and AD exacerbation [3,4].

Therefore, there is an urgent need for new therapeutic targets that can address these challenges. Recent studies have suggested a role for the Hippo signaling pathway and its downstream effector, Yes-associated protein (YAP), in inflammatory skin diseases such as psoriasis. The Hippo signaling pathway is a highly conserved signaling network that controls organ size, tissue homeostasis, and regeneration. Inactivation of the Hippo pathway leads to YAP nuclear translocation, where it functions as a transcriptional co-activator and regulates various target genes involved in cell proliferation, survival, and inflammation [5,6]. Dysregulated YAP activity has been implicated in various human diseases, including cancer, fibrosis, and now inflammatory diseases [7]. Therefore, we sought to investigate the role of YAP in AD, one of the chronic inflammatory diseases.

YAP is typically found in the cytoplasm of epidermal keratinocytes in normal human skin [8]. However, nuclear YAP is observed in areas enriched for epidermal basal cells or somatic stem/progenitor cells [9]. During serial cultivation in vitro, normal human keratinocytes exhibit an increase in nuclear YAP levels. Nuclear YAP is more abundant in early passages, while it is predominantly found in the cytoplasm by later passages [8]. Notably, while YAP expression has yet to be studied explicitly in the skin of patients with AD, *YAP* mRNA has been reported to be significantly increased in Th17 cells from the peripheral blood of individuals with AD [10]. In the skin of individuals with psoriasis, the YAP protein has increased expression, particularly in the stratum basale and lower stratum spinosum layers. Given that the stratum basale is the regenerative area of the epidermis, these findings imply that elevated YAP expression may lead to the excessive proliferation of keratinocytes, directly contributing to the development of psoriasis [11].

YAP lacks a DNA-binding domain; its transcriptional function is driven by its association with other transcriptional factors [12]. In endothelial cells, signal transducer and activator of transcription (STAT)3 could act as a potential DNA-binding domain of YAP [12]. Furthermore, in ulcerative colitis, YAP has been reported to activate the STAT3 signaling pathway in regulating epithelial cell proliferation and promoting mucosal regeneration [13]. Verteporfin acts as a YAP inhibitor by disrupting YAP-TEA domain transcription factor (TEAD) interactions. It is recognized as a porphyrin photosensitizer and is clinically used in photodynamic therapy to treat neovascularization. In the absence of light activation, it is reported to not only inhibit autophagy and induce apoptosis, but also inhibit cell proliferation and differentiation through the regulation of YAP-TEAD within the Hippo pathway [14,15].

In this research, we investigated the differences in the expression of YAP between the skin of AD patients and normal skin, using hapten-induced Nc/Nga mice, an animal model of AD. Nc/Nga mice display a mutation on chromosome 9, which is linked with an increased production of IgE and a heightened Th2 response. Heptane induces the increased expression of IL4 in the dermis and the infiltration of Th2 lymphocytes expressing mast cells and eosinophils into the skin [16].

Therefore, in this study, we explored the role of YAP in AD as well as the network between the JAK–STAT pathway and the Hippo pathway in an AD model using verteporfin, a known YAP inhibitor.

## 2. Results

### 2.1. YAP Is Overexpressed in AD Patients

We investigated the expression of key factors in the Hippo pathway, YAP, TAZ, and large tumor suppressor kinase (LATS)1/2, in AD patients. Immunohistochemistry (IHC) was performed on skin tissues from a total of five patients. YAP expression was observed in epidermal keratinocytes, particularly in the basal-layer keratinocytes, where it was expressed in the cell nuclei (*p* < 0.05). TAZ and P-YAP were weakly expressed in epidermal keratinocytes (Figure 1). LATS1/2 expression was observed in epidermal keratinocytes and dermal fibroblasts. Additionally, TSLP expression was also observed in epidermal keratinocytes and was significantly increased in atopic skin compared to normal skin (*p* < 0.05) (Figure 1).

### 2.2. YAP Inhibitor Attenuates the Symptoms of AD in 1-chloro-2,4-dinitrobenzene-Induced Mice

To investigate the therapeutic effect of YAP inhibitors on AD, inflammation of the dorsal skin of NC/Nga mice was induced using 1-chloro-2,4-dinitrobenzene (DNCB) after a 1-week sensitization period (Figure 2A). The topical application of 0.4% DNCB resulted in AD-like lesions, including edematous erythema with scratching and dryness. DNCB-induced AD mice were treated with vehicle (acetone:olive oil 4:1), 0.03% tacrolimus, 0.02 mM, or 0.2 mM of verteporfin three times per week for a total of 2 weeks. Vehicle was used as an AD control. At the end of the experiment, keratinization, erythema, and edema were observed in the AD control group in contrast to the normal group (*p* < 0.001) (Figure 2B). Also, in the AD control group, AD-like skin inflammation was significantly attenuated in the verteporfin-treated group (*p* < 0.001) (Figure 2B).

Epithelial hyperkeratosis and hyperproliferation, as revealed by hematoxylin and eosin (H&E) staining, were significantly increased in the AD control group compared to the normal group (*p* < 0.001). In addition, both were significantly reduced after treatment with 0.03% tacrolimus (*p* < 0.001), 0.02 mM of verteporfin (*p* < 0.001), and 0.2 mM of verteporfin (*p* < 0.001) (Figure 2C,D). To elucidate mast cell infiltration in the dermis, sections of the dorsal skin were stained with toluidine blue (Figure 2C). As a result, mast cell infiltration and granule production were significantly increased in the AD control group compared to the normal group (*p* < 0.001). In both groups treated with verteporfin, cell infiltration was significantly lower than that in the AD control group (*p* < 0.05) (Figure 2D,E).

### 2.3. YAP Inhibitor Modulates Inflammation-Related Signaling and Improves It at the mRNA Level

To investigate the effect of YAP inhibitors on the development of AD, mouse skin samples were collected and quantified for Th2- and Th1-related signaling genes, which are key factors in the pathogenesis of AD, using quantitative real-time PCR. In this analysis, *Tslp*, *Ifng*, *Il1b*, *Il4*, *Il6*, *Il13*, *Il17*, *Il18*, *Il22*, and *Il33* were significantly increased in the AD control group compared to the normal group (Figure 3). The expression of all cytokines was significantly reduced in the groups treated with 0.02 mM of verteporfin and 0.2 mM of verteporfin compared to the AD control group (Figure 3), except for *Il17* expression in the 0.02 mM of verteporfin-treated group (Figure 3G).

### 2.4. YAP Inhibitors Relieve AD by Blocking Inflammatory Factors and the JAK-STAT Pathway through Inhibition of YAP Protein

We investigated the effects of YAP inhibition in the Hippo pathway on JAK–STAT signaling pathway activation by DNCB in NC/Nga mice. In the AD control group, the expression levels of YAP (*p* < 0.001), TAZ (*p* < 0.001) and P-YAP (*p* < 0.05) were significantly increased while those of YAP (*p* < 0.01) and TAZ (*p* < 0.05) were significantly decreased after treatment with 0.02 mM of verteporfin (Figure 4A). After DNCB-induction in the mouse model, the expression levels of JAK1 (*p* < 0.05), JAK2 (*p* < 0.001), STAT1 (*p* < 0.05), STAT2 (*p* < 0.05), STAT3 (*p* < 0.001), and P-STAT3 (*p* < 0.001) were all increased compared to those in the normal group (Figure 4B,C). TSLP, which plays a key role in Th2 signaling in the early skin inflammatory response, was also significantly reduced by treatment with 0.02 mM and 0.2 mM of verteporfin.

The expressions of JAK1 (*p* < 0.05), JAK2 (*p* < 0.001), STAT1 (*p* < 0.05), STAT2 (*p* < 0.05), and STAT3 (*p* < 0.05) were all downregulated in the presence of YAP inhibition. Indicators of phosphorylation status, such as p-STAT3, tended to decrease in the YAP inhibitor-treated group (*p* < 0.01) (Figure 4B,C).

## 3. Discussion

We identified increased YAP protein levels in the skin of AD patients (Figure 1) and an animal model of AD. In AD animals, we found increased skin thickness and inflammatory cells as well as AD clinical scores. Meanwhile, the YAP inhibitor verteporfin attenuated YAP and reduced TAZ and P-YAP concentrations in the NC/Nga mouse model. Broadly, we observed that the use of YAP inhibitor in an AD animal model significantly reduced clinical scores, skin thickness, and mast cell numbers. In our model, verteporfin also decreased pro-inflammatory, Th1- and Th2-related cytokines. In addition, at the protein level, verteporfin was effective in inhibiting YAP and TAZ expression and reducing levels of TSLP, STAT1, STAT2, STAT3, P-STAT3, JAK1 and JAK2, which are key signaling factors in AD. Despite using concentrations of 0.02 mM and 0.2 mM of verteporfin amidst these circumstances, the results of this study consistently did not show a dose-dependent relationship. However, there was a more pronounced effect at 0.2mM of verteporfin when compared to the placebo control group in terms of clinical scores, skin thickness, and adipocyte count. Significant reductions were observed in the expression of various cytokines, including *Tslp*, *Ifng*, *Il1b*, *Il4*, *Il6*, *Il13*, *Il17*, *Il18*, *Il22*, and *Il33* mRNA, significantly decreased within the lesion tissues. Based on these results, we propose that YAP inhibitors have the potential to alleviate the inflammatory exacerbations in AD.

In AD, the immune response is critical in its pathogenesis and involves a complex network of immune cells and cytokines. Early AD is characterized by a predominant Th2 response, with the Th1 level progressively increasing over time and becoming predominant in the chronic disease phase. TSLP, a key initiator of AD, is expressed in epithelial cells and neurons as well as in TSLP target cells, such as T-cells [17,18,19]. Overexpression of TSLP in mouse skin causes an AD-like disease [18]. In addition, TSLP is known to activate JAK1 through IL-7Rα and JAK2 through TSLPR. JAK1 and JAK2 then activate signal transducers and activators of STAT5A and STAT5B, which in turn activate STAT1 and STAT3, ultimately inducing pro-inflammatory effects as well as IL-4, IL-5, IL-9 and IL-13 [17]. In this study, we identified the expression of TSLP in the skin of AD patients visiting our hospital (Figure 1) and observed that TSLP level was increased by DNCB in mouse skin causing AD-like conditions. In addition, YAP inhibition in the Nc/Nga AD mouse model reduced *Tslp* expression at the mRNA level and significantly reduced the expression levels of both JAK1 and JAK2 and their associated transcription factors STAT1, STAT2 and STAT3.

An increase in secreted cytokines, such as Th2, Th22, Th1, and Th17, is present in AD skin lesions [20,21,22]. We found increased levels of *Il1b*, *Il4*, *Il6*, *Il13*, *Il17*, *Il18*, *Il22*, *Il33*, and Th1 cytokines in the Nc/Nga mouse model of AD. IL-4 is a significant contributor to AD development, inducing Th2 differentiation via the JAK–STAT pathway. We found that *Il4* levels were increased in mouse skin lesions induced by DNCB but decreased in skin treated with a YAP inhibitor. Levels of IFN-γ, which sustains the inflammatory state, IL-6, which contributes to acute phase lymphoid differentiation, and IL-1β, which also encourages Th17 development, were observed in the DNCB-induced AD mouse skin but were all reduced in the YAP inhibitor-induced skin. IL-18, which independently causes AD-like skin inflammation and is associated with severity, and IL-33, an IL-1 cytokine family member that promotes Th2 inflammation and acts upstream in the cascade, were also elevated in the DNCB-induced Nc/Nga AD mouse model [23]. In this study, we found that YAP inhibitors reduced *Il33* and *Il18* levels in mouse skin.

The Hippo signaling pathway contributes to the regulation of cell proliferation and growth. The transcriptional effectors of the Hippo pathway, YAP and TAZ, are expressed in both mouse and human skin, showing a cellular localization pattern in adult human skin [24]. Basal layer cells are characterized by nuclear YAP and TAZ, whereas differentiating daughter cells are characterized by cytoplasmic YAP and TAZ. It has been confirmed that YAP acts as an important regulator of epidermal stem cell proliferation and tissue expansion through interaction with the TEAD transcription factor [25] and plays an important role in developing thick three-dimensional structures [26]. The overexpression of YAP in primary keratinocytes from healthy individuals promotes the immortal proliferation of primary keratinocytes and disrupts the normal differentiation process.

When the Hippo pathway is inhibited, YAP/TAZ is activated and binds to the transcriptional regulator TEAD, which plays an important role in cell cycle and growth, stem cell and regenerative medicine; and in the development, metastasis, drug resistance and recurrence of various cancers [27]. In cancer, inactivation of the Hippo pathway and dysregulation of downstream effectors of YAP and TAZ have been mainly observed; during the aging process, the AMP-activated protein kinase(AMPK) pathway can regulate the activity of the Hippo pathway and the expression of target genes, resulting in reduced cell proliferation and growth [6]. YAP, a key molecule in the Hippo pathway, is overexpressed in several cancers and has shown antitumor effects in combination with verteporfin in neuroblastoma [28], basal cell carcinoma [29] and colorectal cancer [30]. Meanwhile, a recent investigation revealed an upregulation of YAP in Th17 cells within a murine model of AD, although YAP in the epidermis decreased in the early process [10]. Regarding inflammatory diseases, there are reports of reduced vascular endothelial growth factor immunoreactivity when YAP and TAZ are inhibited with verteporfin [31]. Furthermore, it has also been shown that the Hippo pathway plays a key role in both the innate and adaptive immune responses, and YAP/TAZ acts as a mediator of the inflammatory response and is closely associated with some inflammatory pathways like NF-κB and JAK–STAT [7]. We confirmed the activation of YAP/TAZ in an AD model, of which levels reduced upon treatment with a YAP inhibitor. We speculate that the Hippo pathway is linked to a reduction in the activity of inflammatory pathways like JAK–STAT.

Abnormalities in Th2 differentiation and Th2 immune responses, which are important in the initiation and development of AD, are associated with JAK–STAT activation. JAK1 and JAK3 are not only involved in Th1 cell activation during the acute phase of AD, but also exacerbate the inflammatory response in AD by increasing cytokine, chemokine, and IgE production [32,33].

IL-4 type 1 receptors consist of IL-4Rα and γ chains, and IL-4 type 2 receptors consist of IL-4Rα and IL-13Ra1 chains, which also functions as IL-13 receptors. The binding of IL-4 cytokines to type 1 receptors leads to the phosphorylation of JAK1 and JAK3 and activation of STAT6 while binding of IL-4/IL-13 cytokines to type 2 receptors induces the expression of JAK1 and TYK2 and subsequent activation of STAT6 and STAT3 [34]. IL-5 uses JAK1 and JAK2 as well as STAT1, STAT2, and STAT5 [35,36,37]. The intracellular signaling of TSLP and IL-31 is mediated by JAK1 and JAK2, with the subsequent involvement of STAT1, STAT3, and STAT5 [38,39]. Significantly, we identified an increase in *Il4* and *Il13* levels as well as an increase in JAK1/2, STAT1/2/3, and P-STAT3 levels, and a decrease in these after YAP inhibitor treatment in an AD mouse model.

With respect to the JAK–STAT pathway, the role of YAP and TAZ has been reported. YAP 1 knockdown inhibits PD-L1, which is involved in JAK1 and JAK2/STAT1 and STAT3 signaling, and the short TAZ variant, TAZ, has been used to inhibit JAK–STAT signaling and cellular antiviral responses without going through the Hippo pathway [40]. It is also known that activation of the YAP protein depends on gp130 activation, and that the gp130-Src-YAP module, a co-receptor of the cytokine IL-6, is responsible for the activity of YAP [41]. There are also reports of IL-6-gp130–mediated signaling activity in rheumatoid arthritis, leading to JAK1 and STAT3 activity [42,43]. As an independent function of YAP inhibition, verteporfin can induce the cross-linking of signaling proteins to form macromolecular oligomers. This triggered anti-inflammatory effects by inhibiting the NF-κB and JAK/STAT pathways in an inflammatory environment induced by LPS [14]. In this study, we found that both JAK1 and JAK2 and STAT1 and STAT3, which were upregulated in animal models of AD, were downregulated by YAP inhibition. These results suggest that regulation of YAP expression in AD is associated with the JAK–STAT pathway and demonstrates that they act in a network.

Previous studies have indicated that epidermal hyperproliferation is associated with the activation of YAP in an IL-6-dependent manner during inflammatory arthritis. An integral step in IL-6-mediated signaling involves the activation of JAK, a significant cytoplasmic tyrosine kinase constitutively associated with transmembrane gp130. Moreover, previous studies have reported that IL-6 promotes the differentiation of Th17 cells, and IL-17, in turn, enhances IL-6 production in tumors, establishing a positive feedback loop on STAT3 [44]. The use of verteporfin to inhibit epidermal hyperproliferation in an inflammatory environment may be associated with the IL-6/STAT3 pathway [42].

In our study, YAP activation led to an increase in JAK1, JAK2, STAT1, STAT2, STAT3, and P-STAT3 proteins in an animal model of AD. In addition, when treated with YAP inhibitor, the inhibition of JAK–STAT family proteins was consistently shown, and based on the existing literature, YAP does not have a DNA-binding domain, so it performs transcription functions in cooperation with other transcription factors. A direct interaction between YAP/TAZ and STAT3 was demonstrated in transformed ER-Src cells [45]. Therefore, it is inferred that YAP performs the transcriptional transfer process together with STAT as well as TEAD. However, further studies are needed to determine whether YAP directly interacts with JAK/STAT.

## 4. Materials and Methods

### 4.1. Patient Samples

Cutaneous AD samples (n = 5) were collected from patients pathologically diagnosed with AD who visited Seoul St. Mary’s Hospital from 2018–2021. As a control, normal skin samples were obtained from healthy volunteers (n = 3) in the Department of Dermatology of Seoul St. Mary’s Hospital, The Catholic University of Korea. The protocol was approved by the Institutional Review Board of The Catholic University of Korea. (KC16SISI0897) and was performed according to the Helsinki Declaration

### 4.2. Animal Study

Six-week-old male NC/Nga mice (weighing about 16–20 g), were purchased from Japan SLC, Inc. (Shizuoka, Japan). A total of 25 mice were used in the study, with n=5 mice per group. All mice were housed in as 12 h light/dark cycle under as constant temperature of 23 °C ± 3 °C, with humidity of 50% ± 10%. The DNCB (1-chloro-2,4-dinitrobenzene; Sigma-Aldrich, St Louis, MO, USA) solution was prepared at a concentration of 1% and 0.4% in an acetone/olive oil suspension (4:1). For the first week, subjects were sensitized by applying 1% DNCB to their abdomen twice a week. Prior to the challenge phase, the dorsal hair of NC/Nga mice was removed using animal clippers and hair removal cream and allowed to stabilize for 24 h. The challenge phase consisted of applying 0.4% DNCB twice a week for 2 weeks, while also applying vehicle (acetone/olive oil 4:1), 0.02 mM, 0.2 mM of verteporfin (acetone/olive oil 4:1), and 0.03% Tacrolimus ointment, three times a week for the same duration. All procedures of animal research were carried out in accordance with the Laboratory Animals Welfare Act, the Guide for the Care and Use of Laboratory Animals and the Guidelines and Policies for Rodent Experiment provided by the Institutional Animal Care and Use Committee of the School of Medicine, The Catholic University of Korea. (approval no. CUMS-2022-0197-01).

### 4.3. Evaluation of Skin Lesions

The severity of dermatitis was evaluated twice a week after 2 weeks of DNCB treatment. The severities of (1) erythema/hemorrhage, (2) scarring/dryness and (3) edema were scored as 0 (none), 1 (mild), 2 (moderate), and 3 (severe). The dermatitis score was defined as the sum of these individual scores.

### 4.4. Quantitative Real-Time PCR in Dorsal Skin Tissue

Tissue specimens were obtained from the dorsal skin using a 5 mm biopsy punch and RNA was extracted using TRIzol (Invitrogen, Carlsbad, CA, USA). After quantifying the RNA using a NanoDrop (Thermo Fisher Scientific, Waltham, MA, USA), synthesized primers were appropriately diluted and mixed with Power SYBR^®^ Green PCR Master Mix (Takara Biomedical Inc., Shiga, Japan) before conducting real-time PCR analysis using a CFX-96 thermocycler (Bio-Rad, Hercules, CA, USA). The PCR conditions for amplifying all genes were as follows: 10 min at 95 °C, followed by 45 cycles of 95 °C for 15 s and 60 °C for 30 s. Expression data were calculated from the cycle threshold (Ct) values using the ΔCt quantification method. Normalization was performed using the *Actb* expression value. For real-time PCR, the oligonucleotide sequences used were murine β-actin and the following primer sequence (Table 1).

### 4.5. Immunohistochemical Analysis

For immunohistochemical analysis, tissue samples were fixed in 4% formaldehyde, embedded with paraffin wax, and sectioned into 5 μm slices. After deparaffinization and rehydration, antigen retrieval was performed using heated citrate buffer pH6 (Agilent Technologies, Inc., Santa Clara, CA, USA) and the slides were incubated in peroxidase-blocking solution (Agilent Technologies, Inc., Santa Clara, CA, USA) at room temperature for 15 min. The sections were incubated with mouse monoclonal anti-STAT3 (1:300, #9139; Cell Signaling Technology, Danvers, MA, USA), rabbit polyclonal anti-YAP (1:200, sc-15407; Santa Cruz Biotechnology, Dallas, TX, USA), TAZ (1:100, sc-48805), LATS 1/2 (1:200, ab70565; Abcam, Cambridge, UK), phospho-YAP (1:200, #4911) and TSLP (1:100, PA-20320), JAK1 (1:100, #3344), JAK2 (1:1000, #3230), STAT1 (1:2000, #14994), anti-STAT2 (1:400, #72604), anti-P-STAT3 (1:200, #9145) overnight at 4 °C in a humid chamber. Horseradish peroxidase (HRP)-conjugated secondary antibody was detected using the Dako REAL™ EnVision/HRP kit (Agilent Technologies, Inc., Santa Clara, CA, USA) at room temperature for 30 min. Sections were visualized with substrate–chromogen solution for 30 s and then counterstained with Mayer’s hematoxylin (Agilent Technologies, Inc., Santa Clara, CA, USA).

### 4.6. Western Blot

Dorsal skin samples were harvested and stored at −80 °C. Protein lysates were prepared using T-PER lysis buffer (Thermo Fisher Scientific, Waltham, MA, USA) containing a protease inhibitor cocktail (Thermo Fisher Scientific, Waltham, MA, USA). The amounts of protein in the lysates were quantitated using a BCA Protein Assay Kit II (Thermo Fisher Scientific, Waltham, MA, USA). After quantitation, equivalent amounts (20 or 40 μg) of protein were separated on 6–15% SDS-PAGE gels and then transferred to polyvinylidene fluoride membranes (MilliporeSigma, St Louis, MO, USA). After blocking with 5% skim milk or 5% bovine serum albumin (BSA)/Tris-buffered saline with 0.1% of Tween 20 (TBS-T), the samples were incubated overnight at 4 °C with the indicated primary antibodies in 5% BSA/TBS-T. The following primary antibodies were used in the procedures: mouse monoclonal anti–β-actin (1:2500, #3700), STAT3 (1:1000, #9139), YAP (1:1000, sc-376830) and TAZ (1:1000, sc-47724), rabbit monoclonal anti–phospho-YAP (1:1000, #13008), TSLP (1:500, #97630), JAK1 (1:400, #3344), JAK2 (1:1000, #3230), STAT1 (1:1000, #14994), anti-STAT2 (1:1000, #72604), anti-P-STAT3 (1:1000, #9145), and antibodies at 4 °C overnight. The blot membranes were washed 4 times in TBS-T and incubated with horseradish peroxidase-conjugated goat anti-mouse or Rabbit IgG secondary antibodies (GTX213111-01 or GTX213110-01; GeneTex, Irvine, CA, USA) for 2 h at room temperature. After washing 4 times with TBS-T, the bands were visualized by ECL substrate (Thermo Fisher Scientific, Waltham, MA, USA) with an Amersham™ Imager 600 (GE Healthcare, Chicago, IL, USA). The intensity of all observed bands was quantified using Image J software (version 1.8.0) (U.S. NIH, Bethesda, MD, USA).

### 4.7. Digital Analysis of Immunohistochemistry Images

The process of digitizing stained tissue specimens involved the utilization of a DM2500 LED light microscope (Leica Microsystems, Wetzlar, Germany). Quantification of YAP, TAZ, P-YAP, LATS1/2, and TSLP expression levels followed a semi-quantitative protein expression measurement approach, as detailed in reference [46]. This assessment was facilitated through the application of free software ImageJ Fiji (Version 1.54g; WS Rasband, National Institute of Health, Bethesda, MD, USA, http://imagej.net/Fiji/Downloads (accessed on 18 October 2023)).

### 4.8. Statistical Analysis

Statistical analysis was performed by one-way analysis of variance followed by Tukey’s multiple-comparisons test. Unpaired *t*-tests were utilized for comparisons between the two groups. Graphs were generated via GraphPad Prism 5 (GraphPad Software, La Jolla, CA, USA), and all data are presented as mean ± standard error (SEM). Statistical significance was determined when *p* < 0.05 (* *p* < 0.05; ** *p* < 0.01; and *** *p* < 0.001).

## Figures and Tables

**Figure 1 ijms-24-17322-f001:**
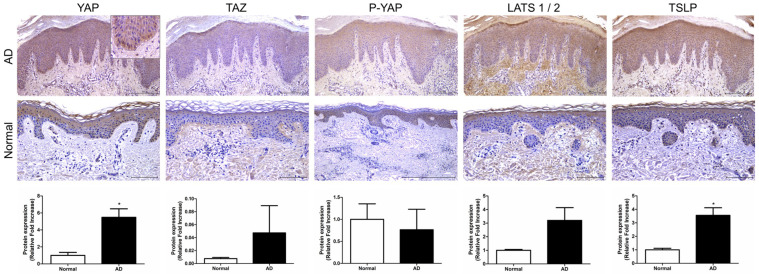
Expression of Hippo pathway component genes in AD patient. Representative image of immunohistochemical staining for YAP, TAZ, P-YAP and LATS1/2 and TSLP in AD and normal skin samples. Original magnification = ×200, scale bars = 100 μm. * *p* < 0.05 compared with normal skin.

**Figure 2 ijms-24-17322-f002:**
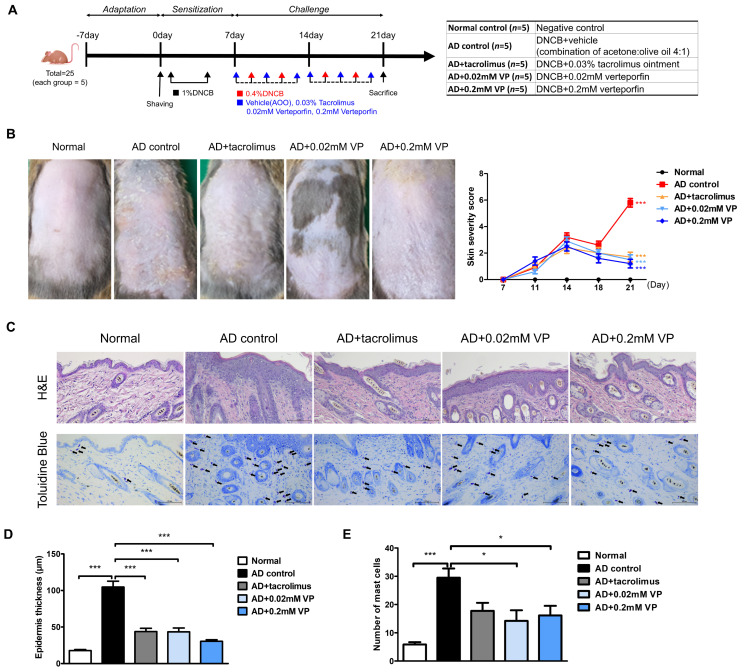
The YAP inhibitor verteporfin reduced skin thickness and mast cell counts in an animal model of AD. (**A**) Time schedule for experimental design. (total *n* = 5/group) (**B**) Clinical results at the end of a 2-week trial in an animal model of AD. Each image shows normal skin, or skin treated with vehicle (acetone: olive oil suspension), 0.03% tacrolimus, 0.02 mM of verteporfin or 0.2 mM of verteporfin on day 21. (**C**) Representative image of H&E and toluidine blue staining in murine skin samples. Histological features of dorsal skin tissue in NC/Nga mice. Original magnification = ×200, scale bar = 100 μm. Mast cells in the dermis are shown in purple. (**D**) Epidermal thickness alteration for each group. (**E**) Mast cell infiltration in the dorsal skin of mice. The results are expressed as means ± standard error values. * *p* < 0.05, *** *p* < 0.001 compared to normal or AD control group.

**Figure 3 ijms-24-17322-f003:**
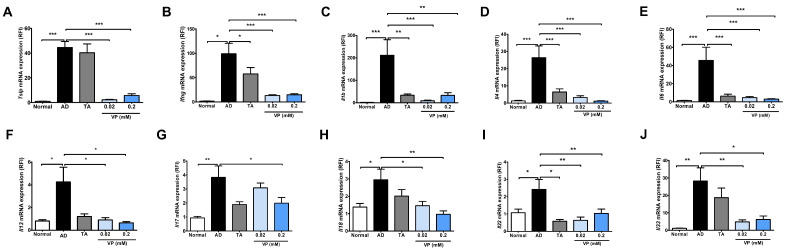
Expression of cytokines after treatment with the YAP inhibitor verteporfin in an animal model of AD. Effects of verteporfin on (**A**) *Tslp*, (**B**) *Ifng*, (**C**) *Il1b*, (**D**) *Il4*, (**E**) *Il6*, (**F**) *Il13*, (**G**) *Il17*, (**H**) *Il18*, (**I**) *Il22*, and (**J**) *Il33* mRNA expression in dorsal skin tissue. Total RNAs were extracted and analyzed by quantitative real-time PCR (total *n* = 5/group). The expression of each gene was normalized to that of *Actb*. Quantitative real-time PCR was performed in duplicate and analyzed by CFX Manager (Bio-Rad Laboratories, Hercules, CA, USA). * *p* < 0.05, ** *p* < 0.01, *** *p* < 0.001 compared to normal or AD group. AD, AD control; TA, 0.03% Tacrolimus; VP, Verteporfin.

**Figure 4 ijms-24-17322-f004:**
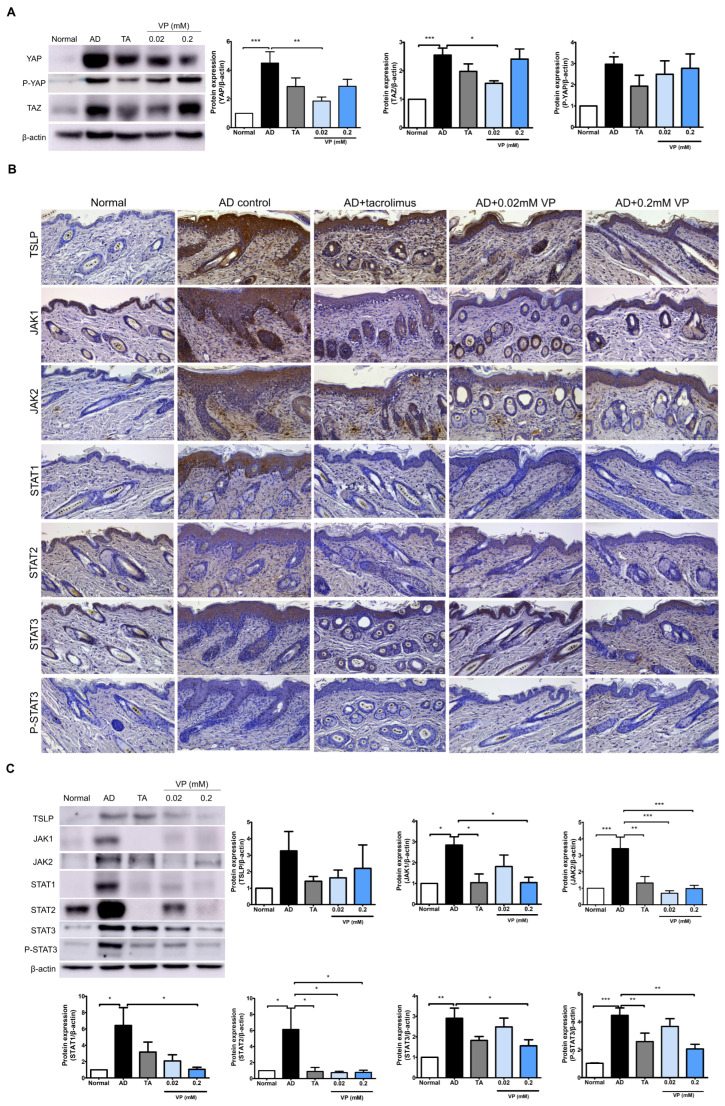
Treatment with verteporfin reduces YAP-associated factors and JAK–STAT protein in an animal model of AD. Differentiation of YAP, P-YAP and TAZ protein expression after verteporfin treatment in an animal model of AD (**A**). Immunostaining JAK–STAT pathway family protein in the same model (**B**). Original magnification, ×200; scale bar = 100 μm. Expression of JAK–STAT pathway family proteins in dorsal skin tissue lysates (**C**). Immunoblotting intensities were calculated with ImageJ software (version 1.8.0). The results are expressed as means ± SEM. * *p* < 0.05, ** *p* < 0.01, *** *p* < 0.001 compared to normal or AD group. AD, AD control; TA, 0.03% Tacrolimus; VP, Verteporfin.

**Table 1 ijms-24-17322-t001:** Primer sequences for quantitative real-time PCR amplifications.

Target	Sequence (5′–3′)	
*Tslp*	Forward	AAAGGGGCTAAGTTCGAGCA
	Reverse	AGGGCTTCTCTTGTTCTCCG
*Il1b*	Forward	TGCCACCTTTTGACAGTGAT
	Reverse	AGTGATACTGCCTGCCTGAA
*Il4*	Forward	TCTCGAATGTACCAGGAGCCATATC
	Reverse	AGCACCTTGGAAGCCTACAGA
*Il6*	Forward	CCCCAATTTCCAATGCTCTCC
	Reverse	AGGCATAACGCACTAGGTTT
*Il13*	Forward	CTGCTACCTCACTGTAGCCT
	Reverse	TATTTCATGGCTGAGGGCTG
*Il17*	Forward	TCCACCGCAATGAAGACCCTGATA
	Reverse	ACCAGCATCTTCTCGACCCTGAAA
*Il18*	Forward	AGGCATCCAGGACAAATCAG
	Reverse	GGTGTACTCATCGTTGTGGG
*Il22*	Forward	CTTGTGCGATCTCTGATGGCT
	Reverse	GCTGGAAGTTGGACACCTCA
*Il33*	Forward	TCCTGTCTGTATTGAGAAACCT
	Reverse	CTTATGGTGAGG CCAGAACG
*Ifng*	Forward	TGATTGCGGGGTTGTATCTG
	Reverse	CTGTCTGGCCTGCTGTTAAA
*Actb*	Forward	TGCTAGGAGCCAGAGCAGTA
	Reverse	AGTGTGACGTTGACATCCGT

## Data Availability

The data presented in this study are available in the article.

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
