# Peer review of "Dysregulated Hippo Signaling Pathway and YAP Activation in Atopic Dermatitis: Insights from Clinical and Animal Studies"

_ijms, 2023, doi:10.3390/ijms242417322_

Round 1

Reviewer 1 Report

Comments and Suggestions for Authors

This is an important paper indicating that YAP is a target for the therapy of AD. There are some issues to be addressed as follows:

Which cell types express YAP in the AD skin lesions? Keratinocytes? That should be explained in introduction.

Please explain the mechanism how verteporfin inhibits YAP.

Why P-YAP expression was not suppressed by verteporfin should be explained.

How does YAP up-regulate jak/stat pathway? Does YAP interact with jak or stat or both?

The effects of verteporfin on stat6, IL-13 , and IL-22 expression should be examined since it suppressed epidermal hyper proliferation and IL-4 expression.

Is the suppression of epidermal hyper proliferation by verteporfin mediated by suppression of IL-6/stat3 pathway? Please present the authors’ opinion.

Did verteporfin reduce NFkB activity in this DNCB-induced AD model? The results should be shown.

Comments on the Quality of English Language

Mostly good.

Author Response

We sincerely thank you for kind reviews and careful reading on our paper. And we appreciate your time and consideration. Please Please see the attachment and find detailed point-by-point response to all comments.

Reviewer 2 Report

Comments and Suggestions for Authors

This study investigated the impact of a YAP inhibitor, Verteporfin, on various cytokines associated with Th1 and Th2 immune responses, and also the components in the JAK-STAT signalling pathways, in the skin of an NC/Nga mouse model. Using several techniques, including histology, mast cell staining, quantitative PCR, and western blotting, the authors showed that Verteporfin had a significant effect on YAP expression, epidermal cell proliferation, the number of mast cells, cytokines, and the JAK-STAT pathway in the skin of mice with atopic dermatitis induced by DNCB. Therefore, the authors suggest that YAP inhibitors could be a potential treatment for atopic dermatitis. However, there are major concerns about the design of the study, the issues of missing data and data presentation, etc. in the current version of the manuscript. It is recommended for authors to address those issues in order to ensure the integrity and validity of the research findings before publication.

Introduction: A brief introduction to provide context about NC/Nga mice and their relevance to the study is necessary.

Figure S1 should be moved to the main body of the manuscript as this is an important part of the study. However, the link provided by the authors (Line 317) is not working so I cannot see the actual data at all.

The major concern from this reviewer is the experimental design of using 0.03% tacrolimus as the normal control throughout the study (Line 64, 'treatment normal group'). This choice lacks a clear justification or citation of relevant literature. Why not consider the placebo control, i.e., animals treated with the vehicle alone (acetone: olive oil 4:1)? This has led me to wonder whether the basal levels of YAP expression in the dorsal skin of mice were high.

Fig. 1: it would be helpful to show a timeline of the experimental design/procedures to allow the readers to see clearly how this experiment was executed. It seems there were no error bars in Fig. 1A graph. Was n=5?

Fig. 1B: The mast cells staining by toluidine blue (purple) cannot be visualised at the magnification shown, as well as those tiny arrows displayed. The enlarged images should be displayed.

Fig. 2: In addition to the qPCR analysis, it is recommended to measure these cytokines in serum using ELISA. This approach allows for the quantification of protein levels, providing a more direct assessment of functional changes, rather than relying solely on messenger RNA (mRNA) expression data.

The statistical methods used to obtain the p values should be described in the figure legends.

2.3: What method was used to analyse the proteins? No indication of the figures was shown in both paragraphs in this section. Additionally, I would recommend the incorporation of IHC to assess the expression of at least some of these proteins, which can provide valuable insights into their localization and distribution within the tissues.

Fig. 3: It is unclear whether the proteins were extracted from the epidermis and there is no description about it. B. it appears that some of the blots may not be fully representative, particularly for proteins like TSLP, STAT1, STAT2, and STAT3. It is crucial to ensure that the presented blots accurately represent the results to maintain the study's scientific rigour and credibility.

The authors employed two concentrations of Verteporfin, i.e. 0.02 mM and 0.2 mM, with the latter being 10-fold higher than the former. However, a dose-dependent effect doesn't appear to have been observed throughout the study. It is advisable for the authors to address this issue explicitly in the final Discussion section, offering potential explanations for the absence of a dose-dependent response. Such an explanation will contribute to a more comprehensive interpretation of the study's findings and their implications.

The negative results for IL-17 are not discussed.

Line 14: repetition

Comments on the Quality of English Language

In general, the manuscript is well-written except for a few minor corrections that are required, e.g. Line 14 'atopic dermatitis' repeated twice, line 224, 't', etc.

Author Response

(The authors gave the same response as above.)

Round 2

Reviewer 1 Report

Comments and Suggestions for Authors

The authors appropriately revised the manuscript.

Comments on the Quality of English Language

No

Author Response

We sincerely thank you for kind reviews and careful reading on our paper. And we appreciate your time and consideration. Please see the attachment.
Thank you.

Reviewer 2 Report

Comments and Suggestions for Authors

Thanks to the authors for addressing my comments. Please see below my responses and additional concerns regarding the current version of the MS:

1) Introduction: A brief introduction to provide context about NC/Nga mice and their relevance to the study is necessary.

A: I totally agreed with the need for a specific introduction regarding the necessity of atopic dermatitis animal models. We have added these sentencesto (Page2 lines67-74) as follows:

This research used hapten-induced Nc/Nga mice as an animal model for atopic dermatitis. Nc/Nga mice display a mutation on chromosome 9, which is linked with an increased production of IgE and a heightened Th2 response. It is well established that heptane induction, causes a dermal infiltration of Tn2 lymphocytes that express mast cells and eosinophils as well as an amplified expression of IL-4 in the dermis referenceJin H, He R, OyoshiM, Geha RS. Animal models of atopic dermatitis. J Invest Dermatol. 2009 Jan;129(1):31-40.

Please re-write this sentence: “It is well established that heptane induction, causes a dermal infiltration of Tn2 lymphocytes that express mast cells and eosinophils as well as an amplified expression of IL-4 in the dermis”

2) Figure S1 should be moved to the main body of the manuscript as this is an important part of the study. However, the link provided by the authors (Line 317) is not working so I cannot see the actual data at all.

A:First, we apologize for the issue with the non-functional link to the 'supplementary figure.' We have now added the content regarding changes in the expression of Hippo pathway components in the skin tissues of atopic dermatitis patients as a primary figure in the main text, and detailed information can be found in Result 2.1.

Unfortunately, there is no data on the normal controls in this figure. As described in 4.1, the authors used 3 control samples for this study and thus, the data for each antibody should be added to the figure.

Line 80: enlarged images with nuclear YAP staining should be displayed.

3) The major concern from this reviewer is the experimental design of using 0.03% tacrolimus as the normal control throughout the study (Line 64, 'treatment normal group'). This choice lacks a clear justification or citation of relevant literature. Why not consider the placebo control, i.e., animals treated with the vehicle alone (acetone: olive oil 4:1)? This has led me to wonder whether the basal levels of YAP expression in the dorsal skin of mice were high.

A: We apologize for any lack of clarity in describing our experimental design. Tacrolimus, an immunomodulator, manages atopic dermatitis by inhibiting T lymphocyte activation, modifying dendritic cell surface expression, and regulating inflammatory mediator release from skin mast cells and basophils. It efficiently penetrates the skin with minimal systemic absorption, maintaining low blood concentrations in clinical trials.The study consisted of two treatment groups: a control group treated with the commonly used topical agent for atopic dermatitis, tacrolimus, and another group treated solely with a vehicle alone (combination of acetone: olive oil 4:1).

All DNCB-induced Nc/Nga mice, except those in the normal group, received one of the following treatments: acetone: olive oil 4:1(AOO), 0.03% tacrolimus, 0.02mM verteporfin, or 0.2mM verteporfin. To improve communication clarity between the AOO-treated and DNCB-induced groups, we have replaced the "DNCB-induced" group with the "vehicle" group. Thank you for understanding.

Unfortunately, to replace the "DNCB-induced" group with the "vehicle" group still does not sound quite correct. Could the authors name the 5 groups of animals with normal control, AD control, AD+tacrolimus, AD+0.02mM and AD+0.2mM VP, respectively, in a table next to Fig. 2A, with the description of different treatments, the number of animals, etc.

4) Fig. 1: it would be helpful to show a timeline of the experimental design/procedures to allow the readers to see clearly how this experiment was executed. It seems there were no error bars in Fig. 1A graph. Was n=5?

A: To help facilitate a better understanding of our experimental process, we have included a timeline in Figure 2A. Additionally, we have reclassified the supplementary figure as Figure 1 in the main text, replacing the previous Figure 1 with Figure 2.method 4.2 animal study

A total of 25 mice were used in the study, with n=5 mice per group.Six-week-old male NC/Nga (weighed about 16–20 g), (Japan SLC, Inc., Shizuoka, Japan) were purchased. A total of 25 mice were used in the study, with n=5 mice per group.All mice were housed in a 12-h light/dark cycle under a constant temperature of 23±3°C, humidity of 50±10%. The DNCB (1-chloro-2,4-dinitrobenzene, Sigma-Aldrich, St Louis, MO, USA) solution was prepared at a concentration of 1% and 0.4% in an acetone: olive oil suspension (4:1). For the first week, subjects were sensitized by applying 1% DNCB to their abdomen twice a week. Prior to the challenge phase, the dorsal hair of NC/Nga mice was removed using an animal clipper and hair removal cream and allowed to stabilize for 24 hours. The challenge phase consisted of applying 0.4% DNCB twice a week for two weeks, while also treating with vehicle (acetone: olive oil 4:1), verteporfin and 0.03% Tacrolimus three times a week for the same duration.

Please refer to my point above and add this info in the table

11) The authors employed two concentrations of Verteporfin, i.e. 0.02 mM and 0.2 mM, with the latter being 10-fold higher than the former. However, a dose-dependent effect doesn't appear to have been observed throughout the study. It is advisable for the authors to address this issue explicitly in the final Discussion section, offering potential explanations for the absence of a dose-dependent response. Such an explanation will contribute to a more comprehensive interpretation of the study's findings and their implications.

A: Unfortunately, this study's results of using 0.02mM and 0.2mM of verteporfin were not consistently dose-dependent. However,there was a more pronounced effect at 0.2mM of verteporfin when compared to the placebo control group in terms of clinical scores, skin thickness, and adipocyte count. Significant reductions were observed in expression of various cytokines, including TSLP, IFN-γ, IL-1β, IL-4, IL-6, IL-13, IL-17, IL-18, and IL-33 mRNA, significantly decreased within the lesion tissues. Importantly, increased expression of JAK1, JAK2, STAT1, STAT2, and STAT3 showed a significant decrease in the 0.2 mM verteporfin treatment group.

Please address the highlight above

The clinical finding is not mentioned in the abstract and also in the last paragraph of the Introduction.

Line 49: this sentence needs to be re-written, ‘In vitro, normal human keratinocytes show higher levels of nuclear YAP in early lineages, shifting to the cytoplasm in later lines[8].‘ Is the YAP expression cell lineage-dependent? Please clarify

2.2: Logically, the authors should confirm the increased YAP expression in the animal before launching the YAP inhibitor study.

Line 94/97: Don’t understand these sentences ‘Meanwhile, 0.03% tacrolimus, a well-known treatment for AD, and vehicle group as a normal control.’ ‘Also, in vehicle group, AD-like skin inflammation was significantly attenuated in the verteporfin-treated group (P < 0.001) (Figure 2B).’ Please re-write

Line 156: “A, total lysates from dorsal skin tissue expressing with JAKSTAT pathway family proteins (C).” What does A mean? Please also double-check this figure legend.

Comments on the Quality of English Language

Some sentences are unclear and need to be rewritten.

Author Response

(The authors gave the same response as above.)

Round 3

Reviewer 2 Report

Comments and Suggestions for Authors

Please address the highlights in the attached file.

Comments on the Quality of English Language

Only minor corrections are required.

Author Response

We thank the reviewer for their careful reading of the manuscript and their constructive remarks. We have taken the comments on board to improve and clarify the manuscript. Please see the attachment.
